# Assessment of dynamic cardiac repolarization and contractility in patients with hypertrophic cardiomyopathy

Andrew E. Radbill[1]*, Lucy Y. Lei[2], Sachin Y. Paranjape[3], Daniel J. Blackwell[3], Robert L. Abraham[4], Derek S. Chew[2¤], Satish R. Raj[2,3], Björn C. Knollmann[3]

1 Division of Cardiology, Department of Pediatrics, Monroe Carell Jr. Children's Hospital, Vanderbilt University Medical Center, Nashville, Tennessee, United States of America, 2 Department of Cardiac Sciences, Libin Cardiovascular Institute of Alberta, University of Calgary, Calgary, Canada, 3 Division of Clinical Pharmacology, Department of Medicine, Vanderbilt University Medical Center, Nashville, Tennessee, United States of America, 4 Division of Cardiovascular Medicine, Arrhythmia Section, Vanderbilt University Medical Center, Nashville, Tennessee, United States of America

¤ Current address: Duke Clinical Research Institute, Duke University Medical Center, Durham, North Carolina, United States of America

* andrew.radbill@vumc.org

**Data Availability Statement:** Data are available from Dryad: https://doi.org/10.5061/dryad.cjsxksn4k.

## Abstract

### Aims

Arrhythmia mechanisms in hypertrophic cardiomyopathy remain uncertain. Preclinical models suggest hypertrophic cardiomyopathy-linked mutations perturb sarcomere length-dependent activation, alter cardiac repolarization in rate-dependent fashion and potentiate triggered electrical activity. This study was designed to assess rate-dependence of clinical surrogates of contractility and repolarization in humans with hypertrophic cardiomyopathy.

### Methods

All participants had a cardiac implantable device capable of atrial pacing. Cases had clinical diagnosis of hypertrophic cardiomyopathy, controls were age-matched. Continuous electrocardiogram and blood pressure were recorded during and immediately after 30 second pacing trains delivered at increasing rates.

### Results

Nine hypertrophic cardiomyopathy patients and 10 controls were enrolled (47% female, median 55 years), with similar baseline QRS duration, QT interval and blood pressure. Median septal thickness in hypertrophic cardiomyopathy patients was 18mm; 33% of hypertrophic cardiomyopathy patients had peak sub-aortic velocity >50mmHg. Ventricular ectopy occurred during or immediately after pacing trains in 4/9 hypertrophic cardiomyopathy patients and 0/10 controls (P = 0.03). During delivery of steady rate pacing across a range of cycle lengths, the QT-RR relationship was not statistically different between HCM and control groups; no differences were seen in subgroup analysis of patients with or without

**Funding:** This work was supported in part by the Vanderbilt Clinical and Translational Science Award grant (AER) from the National Center for Advancing Translational Sciences/National Institutes of Health [UL1 TR000445]. The funders had no role in study design, data collection and analysis, decision to publish, or preparation of the manuscript. There was no additional external or internal funding received for this study.

**Competing interests:** The authors have declared that no competing interests exist.

**Abbreviations:** AV, atrioventricular; HCM, hypertrophic cardiomyopathy; ICD, implantable cardioverter-defibrillator; PVC, premature ventricular contraction.

intact AV node conduction. Similarly, there was no difference between groups in the QT interval of the first post-pause recovery beat after pacing trains. No statistically significant differences were seen in surrogate measures for cardiac contractility.

## Conclusion

Rapid pacing trains triggered ventricular ectopy in hypertrophic cardiomyopathy patients, but not controls. This finding aligns with pre-clinical descriptions of excessive cardiomyocyte calcium loading during rapid pacing, increased post-pause sarcoplasmic reticulum calcium release, and subsequent calcium-triggered activity. Normal contractility at all diastolic intervals argues against clinical significance of altered length-dependent myofilament activation.

## Introduction

Patients with familial hypertrophic cardiomyopathy (HCM) are at risk of sudden cardiac death from malignant ventricular arrhythmias [1]. Although advances in clinical management have reduced the mortality associated with HCM to 0.5% per year [2], the mainstay of current therapeutic approaches for ventricular arrhythmias rely on a "reactive" rather than "preventive" paradigm. An opportunity to prevent ventricular arrhythmias would obviate pitfalls of primary prevention strategies, reduce implantable cardioverter-defibrillator (ICD) treatment failure, eliminate morbidity associated with ICD and myectomy interventions, and increase potential for generalizability across global health care systems. To move towards prevention of ventricular arrhythmias, the mechanisms underlying arrhythmogenesis in HCM hearts must be better understood. Unfortunately, a gap remains between the well-described myocardial pathology and the underlying cellular processes that trigger the electrical events within that milieu [3]. There has been an increasing appreciation for the role of myofilament calcium sensitization in human HCM. Some data suggests this may be a mutation-independent mechanism that alters length-dependent sarcomere activation and thus effects contractility [4]. From an arrhythmia perspective, preclinical mouse models demonstrate increased intracellular calcium buffering producing cardiomyocyte calcium loading during rapid pacing, increased calcium transients following pauses, and subsequent prolongation of the action potential duration, early afterdepolarizations and triggered beats [5]. Increased cytosolic calcium buffering with subsequent alterations in calcium homeostasis and action potentials were also found in human cardiomyocytes derived from induced pluripotent stem cells carrying a pathogenic HCM mutation in troponin T [6]. However, the expression of disease in animal models and humans often varies greatly due to differences in gene dosage, protein isoform background, and post-translational protein modifications [7]. The current pilot study thus sought to explore whether clinical surrogates of contractility and repolarization are altered in humans with HCM in a manner that is concordant with the growing preclinical understanding.

## Materials and methods

### Consent

The Institutional Review Board at both Vanderbilt University Medical Center and the Libin Cardiovascular Institute of Alberta at the University of Calgary approved this research

protocol as compliant with the Declaration of Helsinki. All participants provided written informed consent prior to study participation.

## Study population

The study was conducted at two sites, Vanderbilt University Medical Center and the Libin Cardiovascular Institute of Alberta at the University of Calgary. Potential participants were identified by screening the clinical patient databases of the device and cardiomyopathy clinics. The case group included patients with: 1) clinical diagnosis of hypertrophic cardiomyopathy; 2) age 18–70 years; 3) permanent implanted pacemaker or defibrillator capable of atrial pacing; and 4) left ventricle ejection fraction >50%. Controls were participants who also met inclusion criteria but did not have a diagnosis of hypertrophic cardiomyopathy. Controls were age-matched as a group to the case group (+/-10 years). Exclusion criteria included presence of a cardiac resynchronization device or current use of the following anti-arrhythmic medications: Vaughn-Williams Class I or Class III anti-arrhythmic medications (including disopyramide, sotalol, amiodarone), verapamil or diltiazem. Relevant clinical variables were collected from each participant's medical history and evaluation as detailed below.

## Study protocol

Each participant underwent a single outpatient research visit. The pacing protocol was performed with the participant supine in bed. Each participant was instrumented with EKG electrodes, arm blood pressure cuff, and non-invasive continuous blood pressure recording with a Nexfin device (BMEYE Inc., Amsterdam, The Netherlands). The data were collected digitally onto a laptop computer using an analogue-digital converter for offline analysis. Modelflow™ waveform analysis was used to estimate stroke volume from the pulse contour. Participants underwent a pacing protocol of steady rate drive trains at increasingly faster cycle lengths (857 msec, down to 462 msec) designed to examine the rate-dependence of cardiac repolarization and contractility in humans with HCM. Specifically, using a pacemaker/defibrillator programmer, pacing was delivered at steady rates for 30 seconds, with pauses allowed after drive trains to test whether the post-pause QT-prolongation and ventricular ectopy observed in preclinical HCM models [5] could also be seen in humans with HCM. A 30 second wait was observed between pacing runs to allow for return to baseline. The protocol specified that pacing runs were to be delivered to all patients at the following cycle lengths: 857 msec (70 beats per minute [bpm]), 750 msec (80 bpm), 667 msec (90 bpm), 600 msec (100 bpm), 545 msec (110 bpm), 500 msec (120 bpm), 462 msec (130 bpm). The fastest drive train (462 msec) was repeated 10 times since action potential changes and triggered ectopy were most evident at fast pacing rates in preclinical models. Pacing runs were delivered as atrioventricular (AV) sequential pacing (DDD mode) if 1) the patient's baseline rhythm included ventricular pacing or 2) if there was AV node Wenckebach conduction at atrial rates 462 msec or slower (assessed prior to delivery of pacing runs for all patients with baseline intact AV node conduction). Otherwise, pacing runs were delivered as atrial pacing only (AAI mode). In order to standardize the post-pacing pause duration, the pacemaker lower rate limit was temporarily re-reprogrammed at 1000 msec (60 bpm) at the start of testing; the pacing program was returned to baseline at the conclusion of the research visit. Pacing runs were deferred if the patient's underlying heart rate was higher than the proposed pacing rate. If a patient developed symptoms with drive trains delivered at the faster rates, no additional pacing runs were delivered at that rate or faster; a total of 10 repetitions were performed at the fastest drive train at which the patient remained asymptomatic, as felt appropriate at the discretion of the investigator. Following the final

pacing run, the instrumentation was removed and the patient's participation in the study was complete.

## Statistical analysis

Continuous data are reported as mean with standard deviation. Categorical variables are presented as frequencies with percentages. Demographic and study data were analyzed using t-test for continuous variables and Fisher's exact test for categorical variables. QT was measured at the end of the drive train (penultimate beat in drive train) as well as on the first post pause recovery beat following each drive train. The relationship of cardiac repolarization duration (QT interval) to either pacing drive train rate or post-pacing pause interval was examined by plotting preceding R-R interval (x-axis) against QT interval (y-axis) for each participant and drive train. To account for patient-patient variability and differences in the number of recorded observations, linear mixed-effects models were used to assess the relationship between drive train and QT interval, or post drive train recovery interval and QT interval. Linear mixed-effects models were performed using the lmer() function from the lme4 package in R.

## Results

There were 9 participants in the case group (HCM) and 10 in the control group. Baseline demographics are summarized in Table 1. There was a smaller proportion of females in the HCM cohort as compared to the control group (22% vs. 70%). A similar proportion of participants in each group had AV node disease and required synchronous ventricular pacing (44% HCM, 50% control) during delivery of atrial pacing drive trains (i.e., atrioventricular [AV] sequential pacing). A significantly higher percentage of HCM patients were prescribed maintenance beta-blocker medication (89% vs. 20% in the control group; P = 0.006 by Fisher's exact). Four HCM patients had clinically performed genotyping: *MYBPC3* pathogenic mutation in 3 patients, *MYH7* pathogenic mutation in 1 patient. Regarding the HCM patients' phenotype,

**Table 1. Baseline demographics.**

|  | Hypertrophic Cardiomyopathy (n = 9) | Controls (n = 10) |
|---|---|---|
| Female (%) | 22 | 70 |
| Age (yrs) | 48 ± 12 | 53 ± 12 |
| Ventricular pacing at baseline (%) | 44 | 50 |
| Beta-blocker use (%)[a] | 89 | 20 |
| **Variables specific to hypertrophic cardiomyopathy cohort (n = 9):** | | |
| Interventricular septal thickness (mm) | 18 ± 6 | |
| Sub-aortic peak velocity >50mmHg (%) | 33 | |
| History syncope (%) | 33 | |
| History appropriate ICD therapy (%) | 11 | |
| Genotyping available (%) | 44 (*MYBPC3* x 3, *MYH7* x 1) | |
| **Study day parameters:** | | |
| QRS duration (msec) | 135 ± 37 | 123 ± 36 |
| QT (msec) | 437 ± 28 | 430 ± 37 |
| Cuff blood pressure at start (mmHg) | 115 ± 11/71 ± 9 | 116 ± 9/69 ± 4 |
| Received all protocol-directed drive trains (%) | 67 | 40 |

All continuous data are shown as mean with standard deviation.

[a]Fisher exact 0.006; no other comparisons P<0.05.

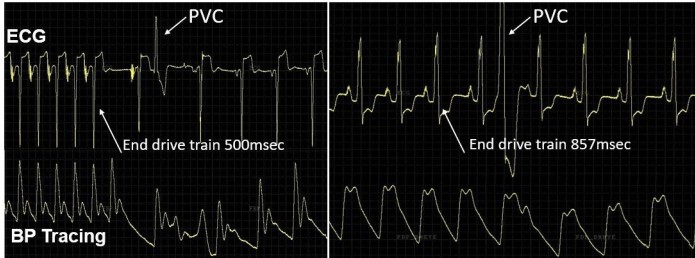

Triggered PVCs: HCM 4/9, Controls 0/10 (P=0.03)

**Fig 1. Following completion of drive train, two examples of triggered ventricular ectopy (premature ventricular contraction [PVC], arrow) in participants with hypertrophic cardiomyopathy.**

mean interventricular septal thickness was 18mm (standard deviation ±6mm), and 3 HCM patients (33%) had a peak sub-aortic velocity of >50mmHg.

The groups had comparable baseline QRS duration, QT interval and cuff blood pressure at the start of testing. A greater proportion of HCM patients (67%) than controls (40%; P = 0.37) completed all protocol-directed drive trains. Three participants in each group did not receive the slowest drive trains (667–857 msec) because the intrinsic atrial rate was greater than the prescribed pacing rate. Three additional control patients displayed variable intrinsic AV node conduction over the range of drive train pacing rates, limiting intra-participant comparisons. For instance, in one participant there was very brisk AV node conduction at baseline and 2nd degree AV Wenckebach conduction at faster pacing rates, precluding comparison of the QRS and QT during slow and fast drive trains. All participants tolerated the first pacing train at the fastest cycle length of 462 msec, although 2 participants did experience transient symptoms limiting the number of repetitions delivered: 1 HCM patient reported transient lightheadedness limiting the number of repetitions delivered at 462 msec, and 1 control described transient chest pain with pacing at 462 msec.

Ventricular ectopy occurred early after pacing drive trains in more HCM patients than controls (44% vs. 0%; P = 0.03) (Fig 1). This ectopy was observed in 4 patients with HCM a total of 5 times, with 4 of 5 occurring following the first post-pacing QRS and 1 of 5 occurring after the sixth post-pacing QRS. In addition, one patient with HCM and post-pacing ventricular ectopy also displayed frequent ventricular ectopy during the drive trains (n = 15); a second HCM patient had a single premature ventricular contraction during a pacing drive train in addition to post-pacing ectopy. None of the 4 HCM patients with triggered PVCs displayed any baseline ventricular ectopy during 5 minutes observation at the start of the study or in the 10 seconds preceding each pacing drive train. There was no difference between HCM patients with ventricular ectopy and those patients without ventricular ectopy (HCM and control) in regard to age, baseline QRS duration, baseline QT interval, baseline blood pressure, sex, ventricular pacing or beta-blocker use. Likewise, no difference in these covariates was seen between HCM patients with and without ventricular ectopy. Additionally, septal wall thickness and peak sub-aortic gradient were similar when comparing HCM patients with and without ventricular ectopy (19 mm vs 17 mm, P = 0.66; 36 mmHg vs 21 mmHg, P = 0.39).

The QT interval was analyzed both during drive train delivery as well as on the first post-pause recovery beat following the drive train. During delivery of steady rate pacing across the range of cycle lengths, the QT-RR relationship was similar amongst HCM and control groups (Fig 2). No differences were seen in subgroup analysis of patients with or without intact AV node conduction (latter group requiring AV sequential pacing rather than atrial pacing only).

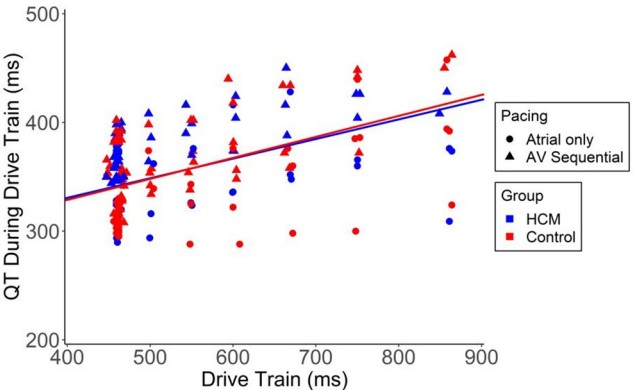

**Fig 2. The QT-RR relationship in control and HCM patients.** Data is shown separately for those patients with intact AV node conduction (atrial pacing only) and those receiving atrioventricular sequential pacing due to absent/inadequate AV node conduction. The QT-RR relationship was similar amongst patients with or without intact AV node conduction in both groups (HCM vs control). Slope for each group: Control = 0.181 [95% CI 0.166, 0.195]; HCM = 0.193 [95% CI 0.177, 0.210]. There were 127 observations from 10 patients in the control group and 74 observations from 8 patients in the HCM group.

Similarly, there was no significant difference between groups in QT interval on the first post-pause recovery beat following delivery of the drive trains (Fig 3).

Regarding surrogate markers of contractility, there was no difference between groups at faster pacing rates in pulse pressure, systolic blood pressure, diastolic blood pressure or calculated stroke volume (Table 2), even when those measurements were normalized to pre-pacing baseline or measurements obtained at slowest pacing rates. Following the fastest tolerated drive trains, controls displayed a non-significant trend towards a greater increase in systolic blood pressure on the post-pacing recovery beat than HCM patients. The only statistically significant difference between cohorts in the contractility analysis included a mildly narrower pulse pressure in the HCM patients at pre-pacing baseline as well as during slowest pacing

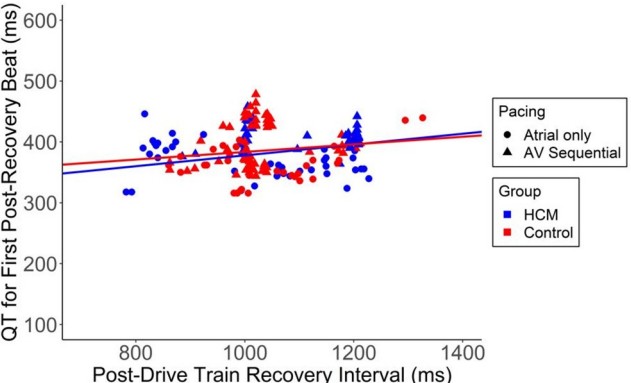

**Fig 3. No significant difference was seen between groups in QT interval on the first post-pause recovery beat following delivery of the drive trains.** Slope for each group: Control = 0.0626 [95% CI 0.0109, 0.115]; HCM = 0.0887 [95% CI 0.0403, 0.136]. There were 127 observations from 10 patients in the control group and 74 patients in the HCM group. Note that in some instances the post-drive train recovery interval was >1000 msec; these instances occurred prior to protocol standardization specifying temporary re-programming of device lower rate limit to 1000 msec.

**Table 2. Summary of contractility assessment during delivery of slowest and fastest drive trains.**

|  | Hypertrophic Cardiomyopathy | Control | P value |
|---|---|---|---|
| **Prior to start of pacing:** |  |  |  |
| *Pulse pressure* (mmHg)[a] | 51 (9) | 66 (10) | 0.01 |
| **Early drive train (5th beat):** |  |  |  |
| *Pulse pressure* (mmHg) |  |  |  |
| Drive train 857 msec[a] | 51 (9) | 65 (10) | 0.02 |
| 750 msec | 51 (12) | 54 (13) | 0.67 |
| 500 msec | 33 (14) | 42 (14) | 0.21 |
| 462 msec | 30 (12) | 35 (10) | 0.4 |
| **Late drive train (penultimate beat):** |  |  |  |
| *Pulse pressure* (mmHg) |  |  |  |
| Drive train 857 msec | 46 (10) | 61 (13) | 0.05 |
| 750 msec | 51 (7) | 57 (13) | 0.25 |
| 500 msec | 36 (12) | 42 (9) | 0.2 |
| 462 msec | 31 (13) | 36 (8) | 0.35 |
| *Stroke volume* (milliliters) |  |  |  |
| Drive train 857 msec | 98 (14) | 86 (16) | 0.23 |
| 750 msec | 93 (12) | 85 (16) | 0.38 |
| 500 msec | 72 (14) | 66 (13) | 0.45 |
| 462 msec | 65 (8) | 58 (12) | 0.2 |
| *Systolic blood pressure* (mmHg) |  |  |  |
| Drive train 857 msec | 114 (11) | 132 (19) | 0.08 |
| 750 msec | 120 (13) | 128 (16) | 0.37 |
| 500 msec | 114 (26) | 118 (10) | 0.71 |
| 462 msec | 107 (12) | 110 (15) | 0.75 |
| *Diastolic blood pressure* (mmHg) |  |  |  |
| Drive train 857 msec | 65 (5) | 70 (9) | 0.28 |
| 750 msec | 68 (6) | 72 (7) | 0.37 |
| 500 msec | 72 (12) | 74 (9) | 0.78 |
| 462 msec | 71 (7) | 72 (8) | 0.86 |
| **Recovery beat after fastest tolerated drive train:** |  |  |  |
| *Change in systolic blood pressure* (mmHg) | +1 (13) | +13 (13) | 0.06 |

Standard deviation shown in parentheses. All comparisons are performed by t-test.

[a]The only statistically significant finding was a mildly narrower pulse pressure in the HCM patients at pre-pacing baseline as well as during slowest pacing rates. Following the fastest tolerated drive train, control participants displayed a non-significant trend towards a greater increase in systolic blood pressure on the post-pacing recovery beat than HCM patients.

rates. Of note, there was also no difference between HCM and control groups in the rate of change in contractility measures following increasingly faster drive trains.

## Discussion

The mechanisms of sudden death in familial hypertrophic cardiomyopathy (HCM) have remained elusive. For this reason, predictive algorithms seeking to identify individuals at high risk of ventricular arrhythmia remain imperfect, and interventions are non-specific and rely on a "reactive" rather than "preventive" paradigm. Early studies identified an association between phenotypic observations of myocyte disarray [8] or ventricular hypertrophy [9], but

the pathophysiology triggering the arrhythmias remained unclear. More recently, there has been a growing number of preclinical studies indicating the role of increased myofilament calcium sensitivity [10]. Although this may be a direct effect of the HCM-causing mutation (such as troponin T mutations), increased myofilament calcium sensitivity may also act independent of the underlying HCM mutation, secondary to reduced phosphorylation of myofilament proteins such as protein kinase A targets [4]. From a contractile perspective, this may alter length-dependent sarcomere activation, with implications for contractility and relaxation [4]. This framework conceptually aligns with the clinical observation of diastolic dysfunction in HCM patients. Regarding arrhythmogenesis, myofilament calcium sensitization in pre-clinical mouse models has been shown to lead to an increase in cytosolic $Ca^{2+}$ binding, prolongation of the action potential duration, and triggered arrhythmia [5]. Here, we sought for the first time to systematically, non-invasively compare dynamic cardiac contractility and repolarization of patients with HCM to that of age-matched controls using a pacing protocol designed to test the effect of increasing pacing rates and pauses.

The most striking finding in our study was that triggered ventricular ectopy was elicited in 44% of HCM patients compared to none of the controls (P = 0.03) during or immediately following the pacing drive trains. In mice carrying an HCM mutation, such ectopy was often observed following the first sinus beat after a pause [5]. Intriguingly, this was also the case in humans with HCM (Fig 1). Based on the experimental work in mice and in human cardiomyocytes derived from induced pluripotent stem cells [6], the underlying mechanism has been ascribed to excessive cardiomyocyte calcium buffering as a result of the increased myofilament calcium sensitivity. This is a near universal finding in HCM regardless of the underlying mutation [11]. In murine models, the enhanced cytosolic calcium buffering leads to cardiomyocyte calcium loading, especially during a fast drive train. During the pause following completion of the drive train, the excess calcium is stored in the sarcoplasmic reticulum and becomes available for release with the subsequent recovery beat. The resulting large calcium transient in the post-pause beat causes excessive action potential prolongation and early after-depolarizations that trigger ventricular ectopy. Accordingly, rapid supraventricular rhythms and increased physical activity have been noted to frequently precede ventricular arrhythmia in HCM patients [12]. Likewise, there has been recent clinical report that atrial fibrillation may predict ventricular arrhythmias in HCM [13].

In vitro studies in HCM animal models and in human myocardium from HCM patients have convincingly demonstrated that the length-dependent activation of the contractile filaments is significantly blunted in HCM regardless of the causative mutation [4]. The *sarcomere length-dependent activation* of force development is the major mechanism responsible for the Frank-Starling effect and hence the beat-to-beat autoregulation of cardiac output [14]. During slow heart rates, the diastolic interval is longer, venous return increased and the end-diastolic sarcomere length increased, enabling a stronger contractile force required to eject the larger blood volume. Given that Frank-Starling mechanism is thought to be impaired in HCM, we had expected to find major differences in contractile measurements at the different pacing rates examined in our study. Surprisingly, no significant differences were identified between HCM and control groups in surrogate non-invasive measures of contractility at any pacing rate, or on the post-pacing recovery beats. The only possible indication for an impaired length dependence in the HCM group was a non-significant trend toward a larger increase in the systolic blood pressure generated by the post-pause beat in the control compared to the HCM group after the last rapid pacing train (Table 2).

Several limitations in this pilot work should be acknowledged. Preclinical work investigating length-dependent activation has demonstrated substantial variance according to the affected gene as well as location and type of mutation in the affected protein [4]. A pilot study

with 9 HCM patients (4 genotyped) could thus be underpowered, particularly in light of the fact that the control population by definition also had non-HCM cardiac disease requiring cardiac implantable device. Also, contractility assessment may be confounded by varying degrees of outflow tract obstruction in the cases, as continuous but non-invasive peripheral blood pressure was used as the primary contractility endpoint. Additionally, the use of beta-blockers in the large majority of HCM patients may confound results: calcium sensitivity and myofilament protein phosphorylation are linked, with the latter driven by beta-adrenergic mediated phosphorylation [6]. Although patients receiving sodium or potassium channel blockers were excluded, this study was not designed to identify or control other factors influencing sodium and potassium currents, with consequent influence on repolarization duration. Females have longer QT interval, and had greater relative representation in the control group (7/10 versus 2/9 in the HCM group). Although this may have made it more difficult to detect significant QT prolongation in our HCM cohort, it should be noted that a post-hoc analysis did not show any effect of sex on the slope of the relationship between post-pacing QT and RR interval. Other possible explanations for the absence of differences between groups include non-uniform distribution of additional factors contributing to QT measurement (left ventricular hypertrophy, autonomic influence), use of pacing rates that were not as rapid as those used in pre-clinical studies, insensitivity of the measurement tools, incomplete delivery of the pacing drive trains, and finally challenges in some patients with ensuring consistent QRS (ventricular depolarization pattern) across all drive trains when variable degrees of intrinsic AV conduction were present. Lastly, there are other potential explanations for the observed triggered ventricular ectopy in the HCM patients which may not be primarily related to calcium handling abnormalities, such as transient cellular ischemia.

## Conclusions

In summary, there has been recent growing preclinical evidence that human HCM myofilaments display calcium hypersensitivity that may be independent of the primary mutation and may underpin much of the clinical pathophysiology. This pilot study compared dynamic cardiac contractility and repolarization in HCM patients to controls both while receiving pacing drive trains as well as immediately afterwards. Triggered ventricular ectopy was elicited in a significant proportion of HCM patients but not in control participants. These findings align with preclinical descriptions in HCM of excessive calcium loading of the sarcoplasmic reticulum during long diastolic intervals, increased post-pause calcium release, and subsequent early afterdepolarizations and triggered activity [5]. The QT-RR relationship was similar amongst HCM and control groups both during pacing drive trains across varying cycle lengths as well as on the post-pacing recovery beat. The finding of normal contractility at all diastolic intervals argues against clinical significance of altered length-dependent myofilament activation in HCM.

## Acknowledgments

The authors would like to acknowledge the assistance of Holly Waldrop, BS for research logistical support and James Leathers, MD, Jessica Delaney, MD and Susan D. Smith, BSN for help with participant enrollment.

## Author Contributions

**Conceptualization:** Andrew E. Radbill, Satish R. Raj, Björn C. Knollmann.

**Data curation:** Andrew E. Radbill.

**Formal analysis:** Andrew E. Radbill, Lucy Y. Lei, Daniel J. Blackwell, Björn C. Knollmann.

**Funding acquisition:** Andrew E. Radbill.

**Investigation:** Andrew E. Radbill, Lucy Y. Lei, Sachin Y. Paranjape, Derek S. Chew, Satish R. Raj.

**Methodology:** Robert L. Abraham.

**Resources:** Robert L. Abraham.

**Supervision:** Björn C. Knollmann.

**Writing – original draft:** Andrew E. Radbill.

**Writing – review & editing:** Derek S. Chew, Satish R. Raj, Björn C. Knollmann.

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
