## [Decision Letter · Decision Letter 0]

26 Nov 2020

PONE-D-20-31816

Assessment of dynamic cardiac repolarization and contractility in patients with hypertrophic cardiomyopathy

PLOS ONE

Dear Dr. Radbill,

Thank you for submitting your manuscript to PLOS ONE. After careful consideration, we feel that it has merit but does not fully meet PLOS ONE’s publication criteria as it currently stands. Therefore, we invite you to submit a revised version of the manuscript that addresses the points raised during the review process.

Please address comments indicated by the Reviewers.

We look forward to receiving your revised manuscript.

Kind regards,

Elena G. Tolkacheva, PhD

Academic Editor

PLOS ONE

Journal Requirements:

Reviewers' comments:

Reviewer's Responses to Questions

**Comments to the Author**

1. Is the manuscript technically sound, and do the data support the conclusions?

Reviewer #1: Yes

Reviewer #2: Partly

Reviewer #3: Yes

2. Has the statistical analysis been performed appropriately and rigorously? 

Reviewer #1: Yes

Reviewer #2: Yes

Reviewer #3: I Don't Know

3. Have the authors made all data underlying the findings in their manuscript fully available?

Reviewer #1: Yes

Reviewer #2: Yes

Reviewer #3: Yes

4. Is the manuscript presented in an intelligible fashion and written in standard English?

Reviewer #1: Yes

Reviewer #2: Yes

Reviewer #3: Yes

5. Review Comments to the Author

Reviewer #1: This is a valuable, well executed study of an important question in hypertrophic cardiomyopathy induced arrhythmias. The findings advance understanding of the role of Ca buffering and the Frank-Starling relation.

Reviewer #2: In this study, the authors investigate the heart rate dependence of repolarization and contraction in patients with HCM compared to normal. This is an interesting study because it is a clinical follow-up to a similar study performed in animals by the same group (ref 5). Moreover, clinical studies are valuable, but difficult to control. The authors are excellent and very capable of performing this study.

An important general concern is that most of the results do not exhibit striking differences. For example, the main results do not reach statistical significance (e.g. Fig 2 and Fig 3). Variability seems high and real physiologically differences seem small. Methods to reduce variability such as including more patients and the suggestions below may be helpful. Finally, there is no difference in contractility between groups.

The analysis of PVC rate dependence is problematic. Including PVCs during the drive train in the analysis is not ideal because these are not well controlled. For example, the number of beats preceding the PVC and, thus, calcium loading will be variable. Furthermore, PVCs during the drive-train will change the drive-train rate and, thus, calcium loading. Since this mostly happened in one patient, the simplest approach may be to remove this patient from the PVC analysis. For the same reasons, including PVCs many beats after the drive train ends (up to 15) is also problematic (lines 148-149). The number and rate of the non-drive train beats preceding the PVC will be unknown and, so will calcium load. Ideally, only PVCs that occur on the first beat after a clean drive-train should be included in the analysis. This is important for any rate dependent phenomena and it may reduce variability in the reported results.

For the data shown in Figure 2, is QT during the drive train included with QT post pause? If so, this may be problematic as it represents two different protocols to measure the rate dependance of repolarization (e.g. standard versus dynamic restitution protocols). This could introduce extra variability. The authors should consider plotting these data separately. What QT interval was used for the drive train – all the beats? The last beat? Or an average of the QT intervals? It should be the last beat or an average. Including QT from multiple beats in a single drive train should be avoided.

For the post pause QT interval data, the description of the results is not very clear. When all drive trains are included, there is no statistical differences. However, when only fast drive trains were included (110-130) there was a statistical difference. This is supposedly shown in Figure 3; however, I don’t see any difference between the groups shown in figure 3. How were the fast rates chosen? When post drive train QTs were normalized to QT drive train, the differences were not significant. In sum, what is the result? It seems to me that drive train had no effect on post pause QT interval.

Regarding the mechanism of ectopy. Could the mechanism of some ectopic beats (e.g. during drive train) be due to triggered activity from DADs? These are pacing and calcium load dependent as well. EADs, on the other hand, are more brady/pause dependent. How were these two mechanisms distinguished? This should be discussed.

Throughout the manuscript, it would be clearer to use bpm or cycle length not both (e.g. figures vs. text).

Could differences in beta-blocker medication between groups explain why some of the results were not different (e.g. contraction). This seems like an important limitation that can significantly affect the results. Acknowledging this, unfortunatly, does not make it go away.

Was HR different between groups? If so, QTc should be used when appropriate.

Reviewer #3: This is a well-written article that describes an important and useful clinical study. Hypertrophic cardiomyopathy is attracting a large amount of attention at the moment and this manuscript adds some interesting observations. The suggestion that mutations (including to MyBP-C and beta myosin heavy chain) are not producing a clinically meaningful change in contractility is interesting, and in full disclosure, surprising to this reviewer.

The only substantive comment relates to the statistical analysis of the regression plots. It is not very clear from the methods how this was performed and more details should be provided. Was ANCOVA used, for example? I am also intrigued by the clustering in the control group for the low drive-train values. Can 2 sets of control patients be distinguished? Finally, given the different number of samples at x value, some points have very high leverage. Was this taken into account?

6. PLOS authors have the option to publish the peer review history of their article (what does this mean?). If published, this will include your full peer review and any attached files.

Reviewer #1: No

Reviewer #2: No

Reviewer #3: No

---

## [Author Response · Author response to Decision Letter 0]

12 Jan 2021

January 10, 2021

Elena G. Tolkacheva, PhD

Dear Dr. Tolkacheva,

Thank you for the thoughtful review of our manuscript entitled "Assessment of Dynamic Cardiac Repolarization and Contractility in Patients with Hypertrophic Cardiomyopathy" (PONE-D-20-31816). We appreciate the comments of the reviewers, and have addressed the questions raised as follows:

Reviewer #2:

The analysis of PVC rate dependence is problematic. Including PVCs during the drive train in the analysis is not ideal because these are not well controlled. For example, the number of beats preceding the PVC and, thus, calcium loading will be variable. Furthermore, PVCs during the drive-train will change the drive-train rate and, thus, calcium loading. Since this mostly happened in one patient, the simplest approach may be to remove this patient from the PVC analysis. For the same reasons, including PVCs many beats after the drive train ends (up to 15) is also problematic (lines 148-149). The number and rate of the non-drive train beats preceding the PVC will be unknown and, so will calcium load. Ideally, only PVCs that occur on the first beat after a clean drive-train should be included in the analysis.

• We have revised the PVC analysis, now limited to only PVCs that occur early after a clean drive-train. This occurred in the same 4 patients with HCM a total of 5 times, with 4 of 5 occurring following the first post-pacing QRS and 1 of 5 occurring after the sixth post-pacing QRS. We did include a qualitative description of the ectopy seen during the drive trains.

For the data shown in Figure 2, is QT during the drive train included with QT post pause? What QT interval was used for the drive train – all the beats? The last beat? Or an average of the QT intervals?

• QT analysis during the drive train included a single measurement at the end of the drive train (penultimate paced beat). The QT post pause was analyzed separately from the QT during the drive train, and is not included in the data shown in Fig 2. This is clarified with additional text in the Methods section.

For the post pause QT interval data, the description of the results is not very clear. When all drive trains are included, there is no statistical differences. However, when only fast drive trains were included (110-130) there was a statistical difference. This is supposedly shown in Figure 3; however, I don’t see any difference between the groups shown in figure 3. How were the fast rates chosen? When post drive train QTs were normalized to QT drive train, the differences were not significant. In sum, what is the result?

• Fig 3 shows the primary analysis of post-pacing QT, with no difference between groups in post-pacing QT following all drive trains across the full range of post-pacing pauses. As described below, we have reanalyzed the data using a linear mixed-effects model that better accommodates the characteristics of this data set. No difference was observed in post-pause QT between groups in this model, and the previously performed sub-analysis has been removed.

Could the mechanism of some ectopic beats (e.g. during drive train) be due to triggered activity from DADs? These are pacing and calcium load dependent as well. EADs, on the other hand, are more brady/pause dependent. How were these two mechanisms distinguished?

• The limitations and confounding variables potentially influencing PVCs that occurred during the drive train have been noted above, and we have thus re-focused the PVC analysis on the observed post-pacing ectopy. This pause-dependent ectopy occurs in a fashion that is most consistent with EADs, which are discussed in the text. 

Throughout the manuscript, it would be clearer to use bpm or cycle length not both.

• This has been standardized to consistently use cycle length in msec.

Was HR different between groups? If so, QTc should be used when appropriate.

• Because all QT measurements were made during pre-specified heart rates (either during steady rate drive trains or on the first post pause recovery beat after drive trains), it was felt to be most appropriate to measure and report uncorrected QT intervals as plotted against the drive train R-R interval or post-drive train recovery interval.

An important general concern is that most of the results do not exhibit striking differences. For example, the main results do not reach statistical significance (e.g. Fig 2 and Fig 3). Variability seems high and real physiologically differences seem small. Methods to reduce variability such as including more patients and the suggestions below may be helpful… Could differences in beta-blocker medication between groups explain why some of the results were not different (e.g. contraction). This seems like an important limitation that can significantly affect the results.

• The authors agree that enrolling more patients and excluding patients on beta-blocker medication may increase ability to identify potential differences between groups. However, the combination of multiple other exclusion criteria (ie, no calcium channel blocker medication) together with the relatively low number of eligible patients willing to participate in a research study constrained recruitment. In post-hoc analysis, the effect of sex on post-pacing QT interval was examined, but had no effect on the slope of the relationship between QT and RR.

Reviewer #3:

It is not very clear from the methods how this was performed and more details should be provided. Was ANCOVA used, for example?

• Thank you for your suggestions. We went back and reanalyzed the data using a linear mixed-effects model. We have updated the methods to clearly state the assumptions and approach. The revised Figure 2 and Figure 3 reflect this new analysis. It should also be pointed out that because no difference was seen in this model between patients with and without intact AV node conduction, the revised Figure 2 now includes data from all patients in each cohort (both those with AV sequential pacing and atrial pacing only, not just AV sequential pacing only). 

I am also intrigued by the clustering in the control group for the low drive-train values. Can 2 sets of control patients be distinguished?

• Our data are indeed clustered by patient. For example, in the original Figure 2, two control patients comprise the upper (>350) cluster and the other two control patients comprise the entirety of the lower (<350) cluster at the fastest drive trains. The analysis using a linear mixed-effects model accounts for the patient-to-patient variability and differences in the number of recorded observations for each individual. Please see separately uploaded response to the reviewers letter for the figure that further illustrates this point (only includes control patients with AV sequential pacing).

Finally, given the different number of samples at x value, some points have very high leverage. Was this taken into account?

• This is now accounted for in the revised model. No leverage is evident on an individual patient basis and the data are homoscedastic. As seen in the reviewer-only figure, the linear mixed-effects model (black dashes) does not skew (previous fit, gray dotted line) away from the patients with missing data at drive train rates > 800 ms.

Thank you for your consideration of this revised manuscript. 

Sincerely,

Andrew E. Radbill, MD

Assistant Professor of Pediatrics

Division of Cardiology

Vanderbilt University Medical Center

---

## [Decision Letter · Decision Letter 1]

26 Jan 2021

Assessment of dynamic cardiac repolarization and contractility in patients with hypertrophic cardiomyopathy

PONE-D-20-31816R1

Dear Dr. Radbill,

We’re pleased to inform you that your manuscript has been judged scientifically suitable for publication and will be formally accepted for publication once it meets all outstanding technical requirements.

Kind regards,

Elena G. Tolkacheva, PhD

Academic Editor

PLOS ONE

Additional Editor Comments (optional):

Reviewers' comments:

Reviewer's Responses to Questions

**Comments to the Author**

1. If the authors have adequately addressed your comments raised in a previous round of review and you feel that this manuscript is now acceptable for publication, you may indicate that here to bypass the “Comments to the Author” section, enter your conflict of interest statement in the “Confidential to Editor” section, and submit your "Accept" recommendation.

Reviewer #2: All comments have been addressed

Reviewer #3: All comments have been addressed

2. Is the manuscript technically sound, and do the data support the conclusions?

Reviewer #2: Yes

Reviewer #3: Yes

3. Has the statistical analysis been performed appropriately and rigorously? 

Reviewer #2: Yes

Reviewer #3: Yes

4. Have the authors made all data underlying the findings in their manuscript fully available?

Reviewer #2: Yes

Reviewer #3: Yes

5. Is the manuscript presented in an intelligible fashion and written in standard English?

Reviewer #2: Yes

Reviewer #3: Yes

6. Review Comments to the Author

Reviewer #2: (No Response)

Reviewer #3: All comments have been addressed. The statistical analysis is improved and linear mixed models seem appropriate for the experimental design.

7. PLOS authors have the option to publish the peer review history of their article (what does this mean?). If published, this will include your full peer review and any attached files.

Reviewer #2: No

Reviewer #3: **Yes: **Kenneth S Campbell

---

## [Editor Report · Acceptance letter]

2 Feb 2021

PONE-D-20-31816R1 

Assessment of dynamic cardiac repolarization and contractility in patients with hypertrophic cardiomyopathy 

Dear Dr. Radbill:

I'm pleased to inform you that your manuscript has been deemed suitable for publication in PLOS ONE. Congratulations! Your manuscript is now with our production department. 

Kind regards, 

on behalf of

Dr. Elena G. Tolkacheva 

Academic Editor

PLOS ONE